# Leveraging Neuron Activation Patterns to Explain and Improve Deep Learning Classifiers

## Abstract

Deep learning models embed all the training information as neuron activation patterns. However, understanding these patterns to improve model performance appears to be a notoriously challenging task. This paper examines the neuron activation patterns of deep learning-based classification models and explores whether model performances can be explained or improved through neurons' activation behaviour. We first show that the entropy of the neuron activation pattern is related to model performance. We then propose a novel modeling approach that given a trained deep learning model, can leverage the neurons' activation probabilities to further boost the classification accuracy. Our comprehensive experimental study shows notable improvements in classification accuracies (sometimes up to 4.7%) on benchmark datasets for both classic fully connected neural networks and advanced convolutional neural networks.

## 1 Introduction

Deep learning allows researchers to engineer better features and representation of data through representation learning (LeCun et al., 2015). Despite the widespread usage of deep learning methods, it is still considered a black box, and there is a lack of understanding of the working procedure of the models (Rai, 2020; Rudin, 2019). Researchers often rely on intuition and domain knowledge when designing deep learning architectures (Shahriari et al., 2015). They use the models' accuracy and loss to evaluate the performance and tune the hyperparameters to optimize the models (Gigante et al., 2019). However, to gain a deeper insight into the model, it is crucial to look beyond its accuracy and loss, i.e., we need to design additional evaluation matrices that can provide a level of confidence in the model predictions and help understand the mechanism of the prediction.

Exploring network architecture can help explain why some models perform better than others and also provide ways of optimizing existing models. For example, by comparing the weight optimization against model architecture, Gaier & Ha (2019) showed that a model's performance depends mostly on its architecture. Zhang et al. (2018) created an explanatory graph that can disentangle different part patterns from feature maps of the convolutional neural network (CNN). In the context of deep learning-based image segmentation, Rahman & Wang (2016) utilized the last layer probability of a pixel belonging to a segmentation category to calculate the intersection over union (IoU) and used it as a loss function to optimize the segmentation model. In semi-supervised learning, the imbalanced distribution of the data among the classes can affect the training performance. This motivated Lai et al. (2022) to propose a robust training process called Smoothed Adaptive Weighting (SAW) that complements the learning difficulty of each class by weighing the weights based on the difficulty and using it in a loss function.

In a deep learning model, the activation of the neurons is the representation of the presence or absence of features in the input data (Zeiler & Fergus, 2014). Such patterns are called neuron activation patterns. In this paper, we study the neuron activation patterns to explain and optimize the deep learning model. Specifically, we focus on improving classification models that are trained on a given dataset to minimize the difference between the model's predictions. The neuron activation pattern that appears during the training contains rich information that, if understood well, can potentially be combined with model prediction to provide a level of confidence in the prediction value. It may allow us to design more efficient models and additional evaluation metrics to help reduce the number of false negatives and false positives. One may also attempt to understand when some

models perform better than others by analyzing the pattern. This motivated us to explore ways to examine neurons' activation patterns, establish their relation to the training accuracy, and propose ways of improving the performance of an existing model.

## 1.1 CONTRIBUTION

We first study the neuron activation pattern using an information-theoretic model. Our proposed metric shows a correlation with the training and testing accuracy on benchmark datasets. Motivated by this, we examine whether neuron activation patterns of a trained model could be leveraged to improve classification further and propose an auxiliary model to achieve better accuracy.

- For the information-theoretic model, we propose a novel method that allows us to measure the predictability of the neurons' activation behaviour leveraging entropy Gray (2011). Our experimental results show that the entropy of the neurons is relevant to the model's performance. 2

- We propose a novel method for improving model performance. Here we use the activation pattern of a trained model to calculate the probability of neurons being activated for different classes and combine it with the model's prediction through an auxiliary neural network. Our results clearly show the effectiveness of our approach in improving classification accuracies of ResNet-50, InceptionV3, and fully connected neural networks for MNIST (2.1%), Fashion MNIST (0.4%), CIFAR-10 (0.6%), CIFAR-100 (1.5%), and Plant Village (2.5%) datasets.

## 2 RELATED WORK

***Neuron Activation Pattern:*** Researchers have investigated both neuron activation patterns and loss functions to explain deep learning models. Olah et al. (2017) studied what the neurons respond to and proposed that neurons work in a group. Li et al. (2017) showed a relationship between the performance of the deep learning model and the convexity of the loss function. Mahendran & Vedaldi (2015) studied the feature maps of the CNN models to understand the activation pattern by inverting the models and showed that there is indeed a relation between the performance of the model and the activation pattern. Kim et al. (2018) vectorized the activation values of hidden layers and created striped patterns to find relevance between pattern and decision. However, they experimented with a small dataset without directions for generalizability.

Bau et al. (2017) studied the relation between the activation pattern and semantic concepts. They optimized the hyperparameters of the deep learning models and quantitively analyzed the effect of changing different parameters. The relevance between activation patterns and semantic concepts was further studied by Fong & Vedaldi (2018). They created vector responses of semantic concepts based on activation patterns and established that the neurons work in a group, and the same neuron can represent multiple concepts. We found a similar characteristic of the neuron activation pattern in our study, where there was an overlap of neurons representing multiple classes. However, with training, a class was represented by more unique neurons.

Gopinath et al. (2019) leveraged the neuron activation pattern to verify inferred formal properties of feed-forward neural networks and studied the relation between the input and output of a neural network to show that a substantial amount of logic of the network is captured in the activation status of the neurons. NEUROSPF, proposed by Usman et al. (2021), is a tool that generates a symbolic representation of deep learning models and allows testing of the model, test input and adversarial example generation, and robustness analysis.

Although there have been several attempts to explain activation patterns, in this paper we take a very different approach to the explanation of the neural network models by using information-theoretic approaches, which allow us to relate entropy-based quantitative measures to the model's performance and help improve the performance of the model.

***Entropy and Deep Learning:*** Entropy is commonly used to quantify errors, e.g., the cross entropy is a widely used loss function Gordon-Rodriguez et al. (2020). Many researchers proposed modifications to this loss function. Martinez & Stiefelhagen (2018) proposed Tamed Cross Entropy, which has the same convergence property as the cross entropy but is more robust against

uniformly distributed label noise, Zhou et al. (2019) proposed Maximum Probability based Cross Entropy (MPCE) loss function, which uses an MPCE based gradient update algorithm and has less back-propagation error than cross-entropy, and Gordon-Rodriguez et al. (2020) proposed to use log-likelihood of the continuous categorical distribution in the place of the cross entropy loss used in label smoothing and actor-mimic reinforcement learning.

Barbiero et al. (2021) proposed an entropy-based linear layer for concept-based deep learning models, which utilizes entropy to choose a limited subset of input concepts, allowing it to provide concise explanations of its predictions. Huo et al. (2020) implemented a maximum entropy regularizer that encourages uniform weight distribution. Li et al. (2020) approximated the gradient of the cross entropy loss function, which is robust against noise and can avoid the vanishing gradient problem.

Although entropy measures have been used for model optimization, we leveraged them to analyze the activation pattern of the models and propose a performance metric that correlates to the model accuracy.

## 3 HYPOTHESIS

We assume that if a neuron is frequently activated for a particular class, then the neuron is representative of that class. Although one neuron can be representative of multiple classes, over the training, we expect the class representatives to have less overlap. Hence, over the training, the neurons are expected to have more certainty in their activation pattern for various classes (rather than being random). Therefore, we expect the 'activation pattern entropy', a measure related to the predictability of a neuron behaviour, as described later in Section 5.1, to decrease. Additionally, if there exist representative neurons of a class, then such neurons should activate more frequently for any data of that class. In particular, we examine the following hypotheses.

**H1:** The entropy of the activation pattern is related to a deep learning model's performance.

**H2:** The activation probabilities of the representative neurons in a model can be leveraged to improve its performance.

## 4 TECHNICAL BACKGROUND

### 4.1 ENTROPY

Entropy is a metric that measures the level of uncertainty in a system, and a rich body of research examines different ways of measuring entropy (Borowska, 2015). Shanon entropy is a widely used metric in information theory (Shannon, 2001), which calculates the average information available based on the probability of a variable's possible outcomes. Let $X$ be a discrete random variable with possible outcomes $x_1, x_2, ..., x_k$ which occurs with probabilities $P(x_1), P(x_2), ..., P(x_k)$, then the Shanon entropy, $H$ of the variable $X$ is defined as follows:

$$H = -\sum_{i=1}^{k} P(x_i) \log P(x_i). \tag{1}$$

A higher value of Shanon's entropy indicates a more uncertain outcome, which is more difficult to predict. In the following section, we leverage entropy to quantify neuron activation frequency.

### 4.2 DEEP LEARNING MODEL AND STABILITY

The purpose of a deep learning model $\phi(s, \alpha) : \mathbb{R}^n \to [0, 1]^K$ with $k$ classes and $\alpha$ parameters is to map each object $s \in S$ from a finite subset of $T$ to a list of $K$ probabilities. The mapping is accomplished by using activation functions thus

$$\phi(s, \alpha) = \rho \circ X(s, \alpha) \tag{2}$$

where $X(s, \alpha) : \mathbb{R}^n \to \mathbb{R}^K$ is the output of the activation of the previous layer and $\rho$ is the activation function defined as $\rho(x) = \rho(x_1, ..., x_k)$. If $\rho$ is the softmax activation function $\rho(x) = \frac{e^{x_K}}{\sum_{k=1}^{K} e^{X_k}}$, then $\rho(x) \in [0, 1]^K$ and $\sum_{i=1}^{K} \rho_i(x) = 1$. Therefore, $\phi(s, \alpha) =$

$\rho \circ X(s, \alpha) = (p_1(s, \alpha), ..., p_k(s, \alpha))$ calculates the probability $p_k(s, \alpha)$ of $s$ belonging to class $k$. The softmax function is order-preserving which ensures that $X_i(s, \alpha) > \max_{j \neq i} X_j(s, \alpha)$, so $p_i(s) > \max_{j \neq i} p_j(s, \alpha)$. Thus, $X(s, \alpha)$ can determine the class of $s$ and it can be written as

$$\delta X(s, \alpha) = X_{i(s)}(s, \alpha) - \max_{j \neq i(s)} X_j(s, \alpha) \tag{3}$$

In Equation 3, if $s$ is classified correctly then $\delta X(s, \alpha) > 0$ which ensures that $X_{i(s)}(s, \alpha)$ is the largest component of $X(s, \alpha)$ and $p_{i(s)}(s, \alpha)$ is the largest probability. Also, $\delta X(s, \alpha) < 0$ indicates that $s$ is classified incorrectly. In a deep learning model with $M$ layers, $X(s, \alpha)$ is a composition of non-linear activation functions that can be represented as

$$X(s, \alpha) = \sigma_M(s, \alpha_M) \circ \sigma_{M-1}(s, \alpha_{M-1}) \circ \ldots \circ \sigma_1(s, \alpha_1). \tag{4}$$

ReLU is a commonly used activation function that can be defined as $\sigma(x_N) = max(0, x_N)$. The training of a deep learning model involves maximization of the accuracy. The training starts by randomly selecting a starting parameter $\alpha(0)$ and over the training using gradient descent algorithms to minimize the loss and maximize the accuracy for parameters $\alpha(n)$. In a deep learning model for $i$th iteration the parameters $\alpha(i)$ are calculated using $\alpha(t)$ for $0 \leq t \leq i - 1$. The accuracy of the model is a percentage of the well-classified elements of the training set and for any time $t$ it can be defined as

$$acc(t) = s \in T : \delta X(s, \alpha(t)) > 0 \tag{5}$$

Berlyand et al. (2021) proved that under certain conditions and for all $\epsilon$, there exists $\delta$ such that if at any time $t_0$ it is $acc(t_0) > 1 - \delta$ then at all later times $acc(t) > 1 - \epsilon$. So, once the accuracy of the deep learning model is sufficiently high, the training converges and the accuracy remains high. In such cases, the models become stable with training. These observations indicate that the entropy could relate to the activation patterns and thus accuracy, which we explore through experimental analysis.

### 4.3 NEURON ACTIVATION PATTERN

In a deep learning model, the decision of each neuron $N$ depends on the neurons in the previous layer (Equation 4). So, the output of the neurons in the last layer is a non-linear combination of the output of the neurons in the previous layers. For an input $s \in S$ the value of a neuron in layer $m \in M$ can be represented as

$$N(s) = \sigma((w_1.N_1(s) + b_1) + \ldots + (w_{m-1}.N_{m-1}(s) + b_{m-1})) \tag{6}$$

Where $w$, $b$, and $\sigma$ are the weight, bias, and ReLU activation functions, respectively. A neuron is considered active if $N(S) > 0$ and inactive if $N(S) \leq 0$. The neuron activation pattern $\beta$ represents the status of activation of a subset of neurons. In the following discussions, $on(beta)$ and $off(beta)$ are the subset of neurons that are active and inactive, respectively. Assume, there are input properties that generate similar output $P$ for a model (i.e., represents the similar class). If such input property exists, then the activation pattern $\beta$ for those inputs should be similar and it can be defined as

$$\beta(S) ::= \bigwedge_{N \in on(\beta)} N(S) > 0 \bigwedge_{N \in off(\beta)} N(S) = 0 \tag{7}$$

In order to identify such input properties, consider activation patterns $\beta$ where for each neuron $N$, all the neurons that feed into $N$ are also included in the pattern and it is represented as $\prec$-$\beta$. Now, for all the neurons in $\prec$-$\beta$ there exist $w$ and $b$ such that $\forall S : \beta(S) \Rightarrow N(S) = \sigma(w.s + b)$. This can be proved for all $N$ by induction over the depth of neurons. Let $N_1, \ldots, N_c$ be all the neurons that feed into $N$. Using Equation 6, we now have

$$N(s) = \sigma((w_1.N_1(s) + b_1) + \ldots + (w_p.N_p(s) + b_p)). \tag{8}$$

By induction, for each $N_i$ ($1 \leq i \leq p$), there exists $w_i$ and $b_i$ such that

$$\forall S : \beta(S) \Rightarrow N_i(S) = \sigma(w_i.s + b_i) \tag{9}$$

For $\prec$-$\beta$, $N_1, \ldots, N_c$ belongs to $\beta$. If we represent the inactive neurons as $N_1, \ldots, N_a$ and the active neurons as $N_a + 1, \ldots, N_p$, then from Equation 7 we get

$$\forall i \in \{1, \ldots, a\} \forall S : \beta S \Rightarrow N_i(S) = 0 \tag{10}$$

$$\forall i \in \{a+1, \ldots, p\} \forall S : \beta S \Rightarrow N_i(S) > 0 \tag{11}$$

From Equations 9 and 11, and from the definition of $\sigma$ one can observe that

$$\forall i \in \{a+1, \ldots, p\} \forall S : \beta S \Rightarrow N_i(S) = \sigma(w_i.s + b_i) \tag{12}$$

Finally, using Equations 6, 10, and 12, we obtain the following implication.

$$\forall S : \beta(S) \Rightarrow N(S) = \sigma(w.s + b) \tag{13}$$

The discussion shows that neurons within the activation pattern are a linear combination of the input and each activation adds linear constraint to the input predicate. So, there exist input properties that are responsible for prediction $P$ for any $S$. It also implies that for $S \in T$ of the same class, the activation pattern will be similar. In (Gopinath et al., 2019) the authors provided a detailed proof of the concept discussed in this section.

Each neuron of a deep learning model is connected to a specific area of the input data through its filters or kernels and biases, and the activation of a neuron reflects the presence or absence of a particular characteristic in that area of the input data (Section 4.3). Through training, a model becomes stable and the accuracy remains high which implies that the prediction of the model becomes more accurate (Section 4.2). As the accuracy of the model improves, it will be better at separating the data of different classes. Thus, the activation pattern of the neurons is expected to become more similar for the data of the same class.

## 5 METHODOLOGY

In this section, we describe the activation pattern entropy and the probability of representative neuron's activation which are at the core of our methodology and experimental design.

### 5.1 ENTROPY OF ACTIVATION PATTERN

Assume that there are $C$ classes in the training dataset, and let $D_i$, where $1 \leq i \leq k$, be the subset corresponding to the $i$th class. Let $\ell$ be a layer in the neural network with $n$ neurons. At the beginning of training, the weights and biases are randomly initialized, and over the training, these values are updated for improved predictions. The neuron activation thus gets more and more influenced by the classes present in the data. Due to random initial weights, the activation of the neurons also becomes unpredictable. In such a case, the activation patterns' entropy is high, reflecting the system's randomness. As the training progresses, the activation pattern is biased by the data, and thus the entropy will decrease.

We now compute the *activation pattern entropy* of a particular neuron, which is based on the idea of measuring the predictability of its activation value. We create a $|D| \times N$ activation pattern matrix, $F$, where $|D|$ is the size of the training dataset, and $N$ is the total number of neurons in the activation layer, except for the neurons in the last layer (output layer). Each entry $(i, j)$ of $F$ contains the activation value of the $j$th neuron subject to the $i$th element of the dataset $D$. We then compute a normalized matrix $F_{norm}$ by dividing each column by the column sum, i.e., $F_{norm}(i,j) = \frac{F(i,j)}{\sum_{i=1}^{|D|} F(i,j)}$.

To examine the predictability of the activation value for $j$th neuron, we categorized its normalized activation values using $R$ equal-size bins. In other words, we create a histogram for the $j$th column values of $F_{norm}(i,j)$. The intuition is that if the neuron's activation is unpredictable, then the histogram will not have well-defined maxima or contain many local maxima. Otherwise, it will be activated for one or only a few classes and likely to produce a global peak. There is an exception, where a neuron may never be activated and will create a global maxima at the bin that contains the 0 value. Therefore, to create the histogram, we only consider the non-zero activation values. Let $B_k$ and $H_k$, where $1 \leq k \leq R$, be the $k$th bin and its number of elements. Let $h_i$ be the normalized value, i.e., $h_i = \frac{H_k}{\sum_{i=1}^{R} H_i}$. We use the vector $\mathbf{F}_v = [h_1, \ldots, h_R]$ to compute the activation pattern entropy $E_j$ for the $j$th neuron. The activation pattern entropy, $E$, of a deep learning model over all the neurons in all the fully connected layers is calculated as $E = \sum_{j=1}^{N} E_j$.

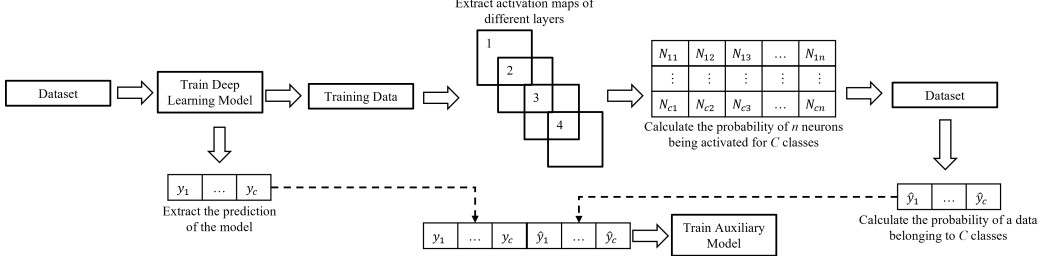

Figure 1: Overview of the proposed performance improvement method and required steps for training the auxiliary model.

## 5.2 PROBABILITY OF NEURON ACTIVATION

In a deep learning classifier, assume there are $c \in C$ classes with $I_c$ members in each class $c \in C$, $n \in N$ layers with $X_n$ neurons in each layer. Let $\sigma_{nx}$ be the activation of $x$th neuron in $n$th layer. We calculate the probability $P_{nc}$ of a neuron $\sigma_{nx}$ being activated for a class $c$ as follows:

$$P_{nc} = \frac{\sum_{i=1}^{I_c} A}{I_c}, \text{ where } A = \begin{cases} 1, \text{ if } \sigma_{nx} > 0 \\ 0, \text{ otherwise} \end{cases} \tag{14}$$

Let $P_{C \times X}$ be the activation probability matrix where each element represents the probability of a neuron being activated for a class. For data with an unknown label, let $U_X$ be the activation probability vector of each neuron. We threshold $U_X$ as follows:

$$U_X^* = \begin{cases} 1, \text{ if } U_X > 0 \\ 0, \text{ otherwise} \end{cases} \tag{15}$$

In a deep learning classifier, Softmax is used at the last layer to generate the likelihood of data being classified into different classes. Let $Y_{predicted}$ be the output of the Softmax activation. So finally, we calculate the probability $Y_{proposed}$ of data being classified into different classes as follows:

$$Y_{proposed} = (P_{c \times X} \cdot U_X^*) \times Y_{predicted} \tag{16}$$

To utilize the proposed probability we train an auxiliary fully connected neural network (FCNN) where the input of the model is the $Y_{predicted}$ and $Y_{proposed}$ and output is the probability of the data being classified into a class. The auxiliary model consists of only 1 hidden layer with neurons twice the number of the classes and ReLU as the activation function.

In Equation 16, $P_{C \times X}$ represents the neuron activation pattern (Section 4.3) of a deep learning model. For a stable classifier (Section 4.2), the probability of activation of the subset of neurons representing a class should be higher than other subsets of neurons. If the base model correctly predicts the class $c$ of a data $s$, then the neuron activation pattern for $s$ which is represented as $U_s^*$, is expected to have the highest overlap with the representative neurons of $c$. Consequently, $(P_{c \times X} \cdot U_X^*)$ will be higher for class $c$ compared to others, and thus $Y_{proposed}$ will also be high for class $c$. Similarly, if $s$ is predicted incorrectly by the base model (i.e., $\arg \max Y_{predicted} \neq c$) and the model is unstable, then $Y_{proposed}$ is expected to be lower for $c$ than other classes. Assume that a base classifier makes a wrong prediction for $s$ despite a stable model and a well-defined neuron activation pattern. If $U_s^*$ shows similar activation as the activation pattern of the class $s$, then the

Table 1: Spearman and Pearson correlation coefficient between the entropy and training accuracy. CIFAR-10 and Plant Village were trained with ResNet-50, CIFAR-100 was trained with InceptionV3, and the rest of the datasets with FCNN. Darker blue represents higher values and darker red represents lower values.

| | MNIST | MNIST Mixed | Fashion MNIST | Fashion MNIST Mixed | CIFAR-10 | CIFAR-100 | Plant Village |
|---|---|---|---|---|---|---|---|
| **PCC** | -0.87 | 0.22 | -0.95 | 0.57 | -0.68 | -0.51 | 0.29 |
| **SCC** | -0.94 | 0.51 | -0.99 | 0.67 | -0.67 | -0.58 | -0.22 |

Table 2: Comparison between a model's Training and Testing accuracy with the proposed Training and Testing accuracy for different datasets. The MNIST and Fashion MNIST were trained using FCNN, CIFAR-10 and Plant Village were trained using ResNet-50, and CIFAR-100 was trained using InceptionV3. The red colour represents the higher value between the model's accuracy and the accuracy of the proposed method. Iter: Iteration, Tr: Training Accuracy (%), PTr: Proposed Training Accuracy (%), Te: Testing Accuracy (%), PTe: Proposed Testing Accuracy (%), Avg.Imp.: Average Improvement (%).

| | MNIST | | | | Fashion MNIST | | | | CIFAR-10 | | | | Plant Village | | | | CIFAR-100 | | | |
|---|---|---|---|---|---|---|---|---|---|---|---|---|---|---|---|---|---|---|---|---|
| Iter | Tr | PTr | Te | PTe | Tr | PTr | Te | PTe | Tr | PTr | Te | PTe | Tr | PTr | Te | PTe | Tr | PTr | Te | PTe |
| 1 | 95.7 | 97.9 | 95.2 | 97.2 | 85.8 | 87.0 | 84.8 | 85.6 | 27.3 | 32.6 | 27.7 | 34.4 | 74.2 | 87.1 | 73.5 | 87.2 | 70.7 | 75.4 | 67.3 | 70.8 |
| 2 | 96.8 | 98.6 | 96.0 | 97.7 | 88.0 | 88.5 | 86.5 | 86.9 | 31.0 | 32.0 | 31.7 | 33.7 | 67.2 | 81.0 | 66.9 | 80.5 | 87.2 | 89.4 | 76.5 | 78.2 |
| 3 | 97.3 | 98.8 | 96.3 | 97.6 | 88.4 | 89.3 | 86.4 | 87.2 | 36.3 | 37.3 | 36.4 | 38.9 | 84.3 | 88.5 | 82.1 | 88.0 | 91.7 | 93.6 | 78.2 | 79.1 |
| 4 | 97.5 | 99.1 | 96.4 | 97.9 | 89.3 | 89.8 | 87.3 | 87.7 | 51.0 | 50.7 | 49.9 | 51.0 | 93.4 | 95.0 | 91.8 | 93.9 | 92.3 | 94.2 | 77.5 | 78.8 |
| 5 | 97.7 | 99.4 | 96.7 | 98.1 | 89.7 | 90.2 | 87.6 | 88.0 | 60.3 | 61.6 | 57.2 | 59.8 | 94.5 | 95.6 | 92.0 | 93.6 | 93.1 | 94.7 | 78.3 | 79.4 |
| 6 | 97.8 | 99.5 | 96.7 | 98.2 | 90.1 | 90.5 | 87.6 | 88.2 | 73.0 | 73.4 | 67.2 | 67.9 | 96.0 | 97.1 | 93.6 | 95.4 | 93.7 | 95.4 | 77.8 | 79.2 |
| 7 | 97.4 | 99.4 | 96.5 | 98.0 | 90.5 | 91.1 | 87.9 | 88.5 | 81.6 | 81.3 | 70.4 | 70.9 | 97.3 | 97.7 | 95.2 | 95.9 | 94.9 | 96.2 | 79.0 | 79.6 |
| 8 | 97.5 | 99.4 | 96.7 | 98.1 | 90.9 | 91.0 | 87.9 | 88.2 | 91.3 | 91.2 | 73.3 | 73.6 | 68.5 | 76.3 | 68.4 | 76.4 | 94.7 | 96.0 | 78.6 | 79.5 |
| 9 | 97.4 | 99.5 | 96.7 | 98.2 | 91.1 | 91.5 | 88.6 | 89.0 | 94.3 | 94.7 | 72.7 | 73.1 | 97.3 | 98.0 | 95.1 | 96.2 | 95.5 | 96.8 | 79.1 | 79.8 |
| 10 | 97.4 | 99.5 | 96.5 | 97.9 | 91.3 | 91.8 | 88.4 | 88.7 | 96.3 | 96.7 | 72.5 | 73.3 | 96.2 | 97.1 | 94.1 | 95.2 | 95.9 | 97.0 | 79.5 | 80.2 |
| 11 | 97.1 | 99.6 | 96.2 | 98.0 | 91.9 | 92.0 | 88.9 | 88.9 | 98.0 | 98.2 | 73.5 | 73.9 | 98.5 | 99.0 | 96.2 | 97.1 | 95.9 | 97.1 | 79.5 | 80.1 |
| 12 | 97.8 | 99.7 | 96.4 | 98.0 | 92.0 | 92.3 | 88.8 | 88.9 | 98.8 | 98.9 | 73.6 | 73.7 | 98.0 | 98.5 | 95.6 | 96.5 | 95.8 | 96.9 | 78.7 | 79.7 |
| 13 | 97.3 | 99.6 | 96.5 | 98.1 | 92.2 | 92.4 | 88.7 | 88.9 | 98.1 | 98.3 | 72.6 | 73.0 | 98.4 | 98.9 | 95.9 | 96.7 | 95.1 | 96.4 | 78.1 | 79.1 |
| 14 | 97.0 | 99.6 | 96.4 | 98.2 | 92.5 | 92.7 | 89.1 | 89.2 | 98.0 | 98.3 | 72.3 | 73.0 | 94.5 | 96.8 | 92.1 | 94.5 | 96.0 | 97.1 | 78.5 | 79.4 |
| 15 | 97.6 | 99.6 | 96.6 | 97.9 | 92.5 | 92.8 | 89.0 | 89.1 | 90.6 | 91.1 | 68.0 | 68.8 | 99.1 | 99.2 | 96.9 | 97.1 | 96.3 | 97.2 | 79.1 | 79.7 |
| 16 | 97.6 | 99.6 | 96.7 | 98.0 | 92.0 | 92.6 | 88.4 | 88.9 | 98.6 | 98.8 | 72.6 | 73.1 | 97.7 | 98.9 | 95.4 | 96.8 | 96.3 | 97.4 | 79.4 | 80.1 |
| 17 | 97.6 | 99.6 | 96.7 | 98.0 | 92.7 | 92.9 | 88.8 | 89.2 | 99.0 | 99.3 | 72.8 | 73.0 | 99.6 | 99.7 | 97.3 | 97.5 | 96.8 | 97.8 | 79.3 | 79.9 |
| 18 | 97.4 | 99.8 | 96.7 | 98.4 | 92.9 | 93.1 | 88.8 | 89.2 | 99.2 | 99.2 | 73.6 | 73.7 | 99.2 | 99.3 | 96.8 | 96.9 | 96.5 | 97.5 | 79.3 | 80.0 |
| 19 | 97.0 | 99.8 | 96.2 | 98.1 | 93.6 | 93.7 | 89.2 | 89.2 | 98.6 | 98.9 | 73.0 | 73.2 | 97.6 | 97.9 | 94.7 | 95.2 | 96.8 | 97.7 | 79.8 | 80.5 |
| 20 | 97.5 | 99.7 | 96.7 | 98.2 | 92.5 | 93.4 | 88.9 | 89.7 | 99.1 | 99.2 | 72.6 | 72.9 | 98.3 | 98.7 | 95.4 | 96.2 | 95.7 | 97.1 | 78.9 | 79.9 |
| Avg. Imp. | 2.1 | | 1.5 | | 0.4 | | 0.4 | | 0.6 | | 1.1 | | 2.5 | | 2.9 | | 1.5 | | 1 | |

proposed probability will be higher than $Y_{predicted}$. Hence, using $Y_{proposed}$ we might be able to improve the prediction.

The benefit of the proposed approach is that we are using an auxiliary model to choose between $Y_{proposed}$ and $Y_{predicted}$. Thus, if $Y_{proposed}$ is incorrect and $Y_{predicted}$ is correct, the auxiliary model will be able to rectify it. In our experiments, we calculated $P_{C \times X}$ using the training data and used it in testing to see the performance. Figure 1 shows an overview of the proposed method. See supplementary material for a detailed overview of the training process.

## 6  DATASET AND MODEL ARCHITECTURE

In our study, we used five benchmark datasets: MNIST (LeCun et al., 2010), Fashion MNIST (Xiao et al., 2017), CIFAR-10 and CIFAR-100 (Krizhevsky et al., 2009), and Plant village (Mohanty, 2018). We also created MNIST Mixed and Fashion MNIST Mixed by randomizing the labels of the respective datasets to examine the reliability of the proposed methods.

In our experiments, we used both FCNN and CNN. See the supplementary materials for the detailed model architecture of FCNN. For CNN, we used ResNet-50, VGG-16, and InceptionV3. For both FCNN and CNN, we collected the weights and bias values from 20 iterations at a uniform interval. Since different models needed a different number of training iterations, this interval length varies, e.g., the CIFAR-10, CIFAR-100 and Plant Village models collected every two iterations.

## 7  RESULT AND DISCUSSION

### 7.1  ENTROPY AND ACCURACY (H1)

A higher entropy value represents more randomness in the activation behaviour of a neuron. At the beginning of the training, due to random weights, the activation of the neuron is random. So, the entropy of the activation pattern should be higher. As the training progresses, the entropy should decrease, representing a biased neuron activation behaviour.

Table 1 shows a negative Pearson (PCC) and Spareman (SCC) correlation between the entropy of activation pattern and training accuracy over model training for most of the datasets. We observed the same relation with the test accuracy. Figure 2 shows the change of normalized entropy with training and testing accuracy for different iterations for different datasets, where a clear relation

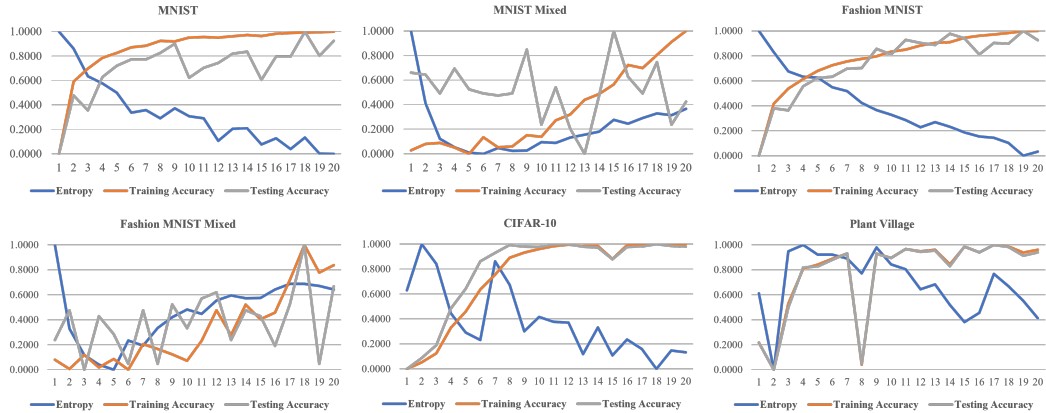

Figure 2: Change of normalized entropy, training and testing accuracy over iterations. MNIST, MNIST Mixed, Fashion MNIST, and Fashion MNIST Mixed were trained on FCNN, and CIFAR-10 and Plant Village were trained using ResNet-50. Since values are normalized, a proper comparison should focus on trends instead of individual peaks or drops.

Table 3: Comparison between a model's Training and Testing accuracy with the proposed Training and Testing accuracy for the CIFAR-10 dataset that was trained using different models. The red color represents the higher value between the model's accuracy and the accuracy of the proposed method. Iter: Iteration, Tr: Training Accuracy (%), PTr: Proposed Training Accuracy (%), Te: Testing Accuracy (%), PTe: Proposed Testing Accuracy (%), Avg.Imp.: Average Improvement (%).

| Iter | FCNN | | | | VGG-16 | | | | ResNet-50 | | | |
|---|---|---|---|---|---|---|---|---|---|---|---|---|
| | Tr | PTr | Te | PTe | Tr | PTr | Te | PTe | Tr | PTr | Te | PTe |
| 1 | 23.1 | 27.8 | 22.9 | 27.3 | 44.5 | 48.1 | 44.0 | 47.7 | 27.3 | 32.9 | 27.7 | 34.2 |
| 2 | 30.7 | 30.5 | 30.4 | 30.8 | 59.8 | 60.0 | 58.7 | 58.7 | 31.0 | 32.3 | 31.7 | 33.8 |
| 3 | 31.6 | 32.2 | 31.0 | 31.9 | 64.8 | 65.5 | 62.4 | 62.8 | 36.3 | 37.2 | 36.4 | 38.7 |
| 4 | 33.6 | 33.3 | 32.8 | 33.4 | 69.0 | 69.6 | 65.8 | 66.2 | 51.0 | 50.9 | 49.9 | 51.3 |
| 5 | 33.1 | 35.4 | 32.0 | 35.1 | 71.7 | 72.4 | 66.7 | 67.9 | 60.3 | 61.7 | 57.2 | 59.8 |
| 6 | 33.2 | 35.8 | 31.9 | 35.3 | 74.5 | 74.9 | 68.8 | 69.3 | 73.0 | 73.3 | 67.2 | 67.9 |
| 7 | 33.9 | 37.6 | 32.3 | 37.0 | 76.4 | 76.5 | 69.7 | 69.8 | 81.6 | 81.5 | 70.4 | 71.0 |
| 8 | 34.3 | 38.4 | 32.6 | 37.3 | 78.1 | 78.6 | 70.3 | 70.9 | 91.3 | 91.2 | 73.3 | 73.6 |
| 9 | 33.5 | 39.0 | 31.5 | 38.3 | 78.7 | 79.5 | 70.2 | 70.8 | 94.3 | 94.7 | 72.7 | 73.1 |
| 10 | 33.8 | 39.8 | 31.6 | 38.2 | 80.8 | 81.2 | 71.0 | 71.5 | 96.3 | 96.6 | 72.5 | 73.2 |
| 11 | 34.6 | 39.9 | 32.1 | 37.9 | 81.3 | 82.2 | 71.1 | 71.8 | 98.0 | 98.1 | 73.5 | 74.0 |
| 12 | 35.1 | 41.1 | 32.2 | 38.7 | 83.3 | 83.5 | 71.3 | 71.4 | 98.8 | 99.0 | 73.6 | 73.7 |
| 13 | 34.7 | 40.8 | 31.6 | 38.9 | 84.4 | 84.6 | 71.6 | 71.9 | 98.1 | 98.3 | 72.6 | 73.1 |
| 14 | 36.9 | 42.0 | 33.5 | 40.3 | 83.8 | 84.7 | 71.2 | 71.6 | 98.0 | 98.2 | 72.3 | 73.0 |
| 15 | 34.5 | 43.1 | 31.8 | 39.7 | 85.2 | 85.8 | 71.7 | 72.0 | 90.6 | 91.2 | 68.0 | 69.0 |
| 16 | 36.0 | 43.6 | 32.4 | 40.2 | 86.8 | 87.0 | 71.9 | 71.8 | 98.6 | 98.9 | 72.6 | 73.0 |
| 17 | 35.4 | 43.3 | 32.0 | 40.2 | 87.1 | 87.5 | 71.4 | 71.9 | 99.0 | 99.3 | 72.8 | 73.1 |
| 18 | 38.4 | 43.6 | 34.2 | 40.5 | 87.4 | 88.0 | 71.3 | 71.9 | 99.2 | 99.2 | 73.6 | 73.7 |
| 19 | 38.5 | 44.7 | 34.6 | 41.4 | 88.7 | 89.1 | 71.2 | 71.5 | 98.6 | 98.9 | 73.0 | 73.2 |
| 20 | 37.6 | 45.0 | 33.0 | 41.6 | 88.9 | 89.6 | 71.8 | 71.7 | 99.1 | 99.2 | 72.6 | 72.8 |
| Avg. Imp. | 4.7 | | 5.4 | | 0.6 | | 0.6 | | 0.6 | | 1.1 | |

between the entropy and model performance can be observed for most of the datasets. However, MNIST Mixed and Fashion MNIST Mixed have positive correlation values, indicating the absence of activation pattern with training due to its random labels, which is consistent with hypothesis **H1**. Although, the Plant Village has a positive PCC, the trend of the entropy is decreasing with training. See the supplementary materials for a detailed overview of the change of entropy for different classes.

## 7.2 IMPROVING MODEL PERFORMANCE (H2)

The discussion of **H2** shows that the representative neurons for each class are highly probable to be activated for the data of that class. We leverage this to improve the deep learning model's performance. Table 2 shows a comparison between the accuracies of the trained and proposed auxiliary model. Our method can be applied to any trained model and it only depends on the activation values of that training instance. As we can see from Table 2, our proposed method helps to increase the accuracy of a model at any given iteration.

Table 4: Comparison between a model's Training and Testing accuracy with the proposed Training and Testing accuracy for the CIFAR-10 and Plant Village dataset that was trained using ResNet-50. The red colour represents the higher value among the model's accuracy, the accuracy of the proposed method using the last 2 activation layers, and the accuracy of the proposed method using the last activation layer. Iter: Iteration, Tr: Training Accuracy (%), PTr-L2L: Proposed Training-Last 2 Layers Accuracy (%), Ptr-LL: Proposed Training-Last Layer Accuracy (%), Te: Testing Accuracy (%), PTe-L2L: Proposed Testing-Last 2 Layers Accuracy (%), PTe-LL: Proposed Testing-Last Layer Accuracy (%), Avg.Imp.: Average Improvement (%).

| | CIFAR-10 | | | | | | Plant Village | | | | | |
|---|---|---|---|---|---|---|---|---|---|---|---|---|
| Iter | Tr | PTr-L2L | PTr-LL | Te | PTe-L2L | PTe-LL | Tr | PTr-L2L | PTr-LL | Te | PTe-L2L | PTe-LL |
| 1 | 27.3 | 32.6 | 32.9 | 27.7 | 34.4 | 34.2 | 74.2 | 87.1 | 87.0 | 73.5 | 87.2 | 87.1 |
| 2 | 31.0 | 32.0 | 32.3 | 31.7 | 33.7 | 33.8 | 67.2 | 81.0 | 80.6 | 66.9 | 80.5 | 80.6 |
| 3 | 36.3 | 37.3 | 37.2 | 36.4 | 38.9 | 38.7 | 84.3 | 88.5 | 88.5 | 82.1 | 88.0 | 87.8 |
| 4 | 51.0 | 50.7 | 50.9 | 49.9 | 51.0 | 51.3 | 93.4 | 95.0 | 95.2 | 91.8 | 93.9 | 93.9 |
| 5 | 60.3 | 61.6 | 61.7 | 57.2 | 59.8 | 59.8 | 94.5 | 95.6 | 95.6 | 92.0 | 93.6 | 93.7 |
| 6 | 73.0 | 73.4 | 73.3 | 67.2 | 67.9 | 67.9 | 96.0 | 97.1 | 97.0 | 93.6 | 95.4 | 95.4 |
| 7 | 81.6 | 81.3 | 81.5 | 70.4 | 70.9 | 71.0 | 97.3 | 97.7 | 97.7 | 95.2 | 95.9 | 95.8 |
| 8 | 91.3 | 91.2 | 91.2 | 73.3 | 73.6 | 73.6 | 68.5 | 76.3 | 75.9 | 68.4 | 76.4 | 76.0 |
| 9 | 94.3 | 94.7 | 94.7 | 72.7 | 73.1 | 73.1 | 97.3 | 98.0 | 98.1 | 95.1 | 96.2 | 96.2 |
| 10 | 96.3 | 96.7 | 96.6 | 72.5 | 73.3 | 73.2 | 96.2 | 97.1 | 97.1 | 94.1 | 95.2 | 95.0 |
| 11 | 98.0 | 98.2 | 98.1 | 73.5 | 73.9 | 74.0 | 98.5 | 99.0 | 98.9 | 96.2 | 97.1 | 96.9 |
| 12 | 98.8 | 98.9 | 99.0 | 73.6 | 73.7 | 73.7 | 98.0 | 98.5 | 98.5 | 95.6 | 96.5 | 96.5 |
| 13 | 98.1 | 98.3 | 98.3 | 72.6 | 73.0 | 73.1 | 98.4 | 98.9 | 99.0 | 95.9 | 96.7 | 96.8 |
| 14 | 98.0 | 98.3 | 98.2 | 72.3 | 73.0 | 73.0 | 94.5 | 96.8 | 96.8 | 92.1 | 94.5 | 94.5 |
| 15 | 90.6 | 91.1 | 91.2 | 68.0 | 68.8 | 69.0 | 99.1 | 99.2 | 99.2 | 96.9 | 97.1 | 97.1 |
| 16 | 98.6 | 98.8 | 98.9 | 72.6 | 73.1 | 73.0 | 97.7 | 98.9 | 98.9 | 95.4 | 96.8 | 96.8 |
| 17 | 99.0 | 99.3 | 99.3 | 72.8 | 73.0 | 73.1 | 99.6 | 99.7 | 99.6 | 97.3 | 97.5 | 97.4 |
| 18 | 99.2 | 99.2 | 99.2 | 73.6 | 73.7 | 73.7 | 99.2 | 99.3 | 99.4 | 96.8 | 96.9 | 96.9 |
| 19 | 98.6 | 98.9 | 98.9 | 73.0 | 73.9 | 73.2 | 97.6 | 97.9 | 97.9 | 94.7 | 95.2 | 95.2 |
| 20 | 99.1 | 99.2 | 99.2 | 72.6 | 72.9 | 72.8 | 98.3 | 98.7 | 98.7 | 95.4 | 96.2 | 96.3 |
| Avg. Imp. | | 0.6 | 0.6 | | 1.1 | 1.1 | | 2.5 | 2.5 | | 2.9 | 2.8 |
| PCC | | 1.0 | | | 0.99 | | | 0.99 | | | 0.99 | |

Next, we investigate whether the proposed method can be used for CNN and FCNN. We trained the CIFAR-10 dataset on FCNN, VGG-16, and ResNet-50. Table 3 shows the comparison between the model's accuracy and accuracy acquired through our proposed method. We were able to achieve improved accuracy for all three models. Layers at a higher depth in a deep learning model are expected to learn the features extracted in the previous layers (Zhang et al., 2018; Zeiler & Fergus, 2014; Simonyan et al., 2013). So, instead of using the activation maps of all the layers, we can use the activation maps in the layers at higher depths to analyze the activation pattern. To investigate this, in Table 4 we compare the accuracies of the trained model and the proposed auxiliary model by using the last 2 activation layers and only the last activation layer. In Table 4, a high PCC value indicates that comparable results can be achieved by only using the activations of the last layer.

The results in Table 2, Table 3, and Table 4 show that using the proposed technique, we can improve the performance of a trained deep learning model, and thus support **H2**.

# 8 CONCLUSION AND FUTURE DIRECTIONS

We have proposed novel methods to explain the behaviour of neurons in deep learning models and leveraged the neuron activation probabilities to improve the classification accuracy. For the entropy-based approach, we found entropy to show a negative correlation with the model performance. Our experimental results showed that the proposed auxiliary model improves the accuracy of many commonly used deep learning models on benchmark datasets.

There are many scopes for future research. Although we examine a diverse set of datasets, adding more datasets could strengthen the results. While we used entropy, there is still scope for designing better quality metrics. Another exciting direction of study can be to use the entropy as a regularization in the loss function Leavitt & Morcos (2020) to better optimize the performance of the deep learning models. We believe that our results will inspire further research to explain the deep learning models using information-theoretic methods.

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
