# OpenReview forum: "Leveraging Neuron Activation Patterns to Explain and Improve Deep Learning Classifiers"
_ICLR.cc/2024/Conference — ICLR 2024 Conference Withdrawn Submission_

### Official Review · Reviewer_s1Ur · 2023-10-15

**Soundness:** 4 excellent
**Presentation:** 4 excellent
**Contribution:** 1 poor
**Rating:** 3
**Confidence:** 2

**Summary:**

This paper proposes using interpretability tools in a logical, sound, and insightful manner to understand how deep learning models work. This is then used to improve the predictive performance of deep learning models. Convincing and sufficient validation is presented for the proposed approach.

**Strengths:**

The paper is technically sound, impactful, meaningful, and opens up brilliant new directions using tools typically reserved for interpretability and explainability. The reviewer believes that this work has significant insight and realizes that these tools can be used for interpreting behavior of neural networks, and in a reinforcement learning loop to continue to improve the performance and predictive power of deep learning models.

**Weaknesses:**

This paper is not well aligned with recent works and overall research direction in the interpretability, explainability, ethics, and the usage of such techniques in a humanist, and ethical manner.

Finalize Doshi-Velez, and Timnit Gebru have written extensively in academic, and non-academic environments about the growing problem with AI models being overly large, and overly powerful. This pattern and trend, and the ownership of these models by private corporations can lead to significant harm.

**Questions:**

I have no questions for the authors.

**Details Of Ethics Concerns:**

I find this paper uncomfortable to digest and accept given recent outcry in the AI communities regarding the growing power of AI models. I think research along this direction and accepting research in this direction puts ICLR, reviewers, PCs, and SPCs in a significant moral quandry.

---

### Official Review · Reviewer_feHw · 2023-10-30

**Soundness:** 3 good
**Presentation:** 3 good
**Contribution:** 2 fair
**Rating:** 5
**Confidence:** 4

**Summary:**

This paper focuses on exploring neuron activation patterns to improve performance of deep learning classifiers. The authors argue that the entropy of the neuron activation pattern is related to model performance. The authors utilize the neurons’ activation probabilities to help boosting the predicted values of base model. The experimental results prove the validity of the theoretical hypothesis and the proposed method.

**Strengths:**

1.	This paper discusses the definition of neuronal activation patterns in detail. Assumptions and theories are available in the paper, and the narrative logic is strong.
2.	In this paper, we hope to directly constrain the activation pattern of neurons to enhance the final classification performance.

**Weaknesses:**

1.	The auxiliary model has some neurons, and the classification tasks and datasets are simple. I suspect that part of the performance gain is due to the extra parameters in this part.
2.	It seems that an auxiliary model is not needed, and the probability of neuronal activation may serve as a new loss function.
3.	In the author's hypothesis, specific neurons are expected to be activated frequently for a particular class. However, in the proposed process of calculating the activation probability of neurons, neurons may be activated by multiple classes, and serve the calculation of activation probability of multiple classes.
4.	The notation in Figure 1 should be consistent with the description in Section 5.2 to avoid confusion to the reader. In particular, the input and output of the auxiliary model should be clearer.

**Questions:**

Please see the weakness.

---

### Official Review · Reviewer_YNcH · 2023-11-02

**Soundness:** 3 good
**Presentation:** 3 good
**Contribution:** 3 good
**Rating:** 6
**Confidence:** 4

**Summary:**

This paper analyzes neuron activation patterns in deep learning models to see if model performance can be improved. The authors find activation entropy is negatively correlated with model accuracy. They propose a novel approach leveraging neuron activation probabilities to boost classification performance of trained models, demonstrating notable accuracy improvements on benchmark datasets. The proposed techniques help explain neuron behavior and use activation probabilities to enhance common deep learning models. Key findings show activation entropy relates to model performance, and the proposed auxiliary model improves accuracy across many datasets. The research provides new methods to explain and improve deep learning through information-theoretic analysis of neuron activations.

**Strengths:**

1. This paper works on an important and timely topic.
2. The proposed methods provide novel ways to explain neuron behavior in deep learning models by analyzing activation patterns. More importantly, it leverages neuron activation probabilities to improve classification accuracy of trained deep learning models across several benchmark datasets.
3. Experimental results show that the proposed method improves accuracy across many common deep learning models. This research provides new techniques to explain and enhance deep learning through information-theoretic analysis of neuron activations.

**Weaknesses:**

Refer to the Questions section

**Questions:**

1. Table 2 shows that the improvement is higher during the earlier stage of model training, and gradually decreases along with the model training. It is interesting to know whether the Te and PTe performance could converge as the Iter continues to increase, e.g., when Iter equals 40.
2. This work is based on the strong hypothesis that: H1: The entropy of the activation pattern is related to a deep learning model’s performance. H2: The activation probabilities of the representative neurons in a model can be leveraged to improve its performance. It would be great if the authors could give some hints when these two hypotheses hold and when these two hypotheses do not hold.

---

### Official Review · Reviewer_zAME · 2023-11-05

**Soundness:** 1 poor
**Presentation:** 2 fair
**Contribution:** 2 fair
**Rating:** 1
**Confidence:** 4

**Summary:**

The paper explores probabilities of activations as a means to improve accuracies on a test set. In Section 5.2 they use a class-probability reweighted prediction, and input this prediction together with the original prediction into a 3 layer neural network to arrive at the final prediction.

**Strengths:**

They perform experiments with an MLP head on top of neural network outputs.

**Weaknesses:**

Certain issues:

issue 1:

Section 5.2  makes no use of the entropy from section 5.1. The whole entropy discussion is not used for the prediction.

equation 16 computes as modified prediction, the prediction for a class weighted by the probabilities for this class that neurons get activated. Effectively it is upweighted by probabilities of neurons being activated.
This is very likely not improving test time prediction on a wide range of networks.

Looking into the code, it becomes clear that the effect comes from putting an MLP head on top of the networks. This has nothing to do with probabilities of activations or entropies.

The effect comes from the MLP head. Putting some reweighted outputs into an additional MLP has very little novelty.

issue 2: The paper has no explaining aspect in it.

issue 3: Aside from that there are lots of mistakes in the intro:

eq 4 is wrong: every concatenation references the input s.
They meant to concat the layers.
\sigma_{M} there is either undefined, or references the sigma in the line below, in which case it would be not correct


In a deep learning model for
ith iteration the parameters α(i) are calculated using α(t) for 0 ≤ t ≤ i − 1.

- this sentence contains no information!

eq 5 is not an accuracy because it outputs a set, not even a count. There is little point defining trivialities like accuracy.


eq 7 is a boolean statement which evaluates to true or false.

"In the following discussions, on(beta) and of f (beta)
are the subset of neurons that are active and inactive, respectively."

Therefore eq.7 is a tautology, as it always evaluates to True .

Something is formally wrong in eq 9 when they want to describe an activation pattern.
It says True implies a that a neuron is an affine mapping with a non-linearity. This is an assumption and has nothing to do with mathematical induction.


equation 13 states the same natural assumption again


"The discussion shows that neurons within the activation pattern are a linear combination of the input
and each activation adds linear constraint to the input predicate."

The first is a trivial assumption. The second is unclear or trivial.

So, there exist input properties that
are responsible for prediction P for any S.

A trivially true statement.

It also implies that for S ∈ T of the same class, the
activation pattern will be similar

In this generality a true statement.

5.2: U_X is positive whenever P_{nc} is positive. Therefore U_X^* in eq 16 adds no information.

**Questions:**

In a deep learning classifier, assume there are c ∈ C classes with Ic members in each class c ∈ C, - how is this computed for test data ?

---

### Author Response · Authors · 2023-11-21
**Withdrawal of Submission for Further Study: Acknowledging Reviewer Feedback**

We extend our gratitude for the insightful feedback provided by the reviewer. After thorough deliberation among the paper's authors, it has become evident that additional study within the proposed research domain is essential. Consequently, we have made the decision to withdraw the submission.